# Prevalence and Clinical Impact of Coinfection in Patients with Coronavirus Disease 2019 in Korea

**DOI:** 10.3390/v14020446

**Published:** 2022-02-21

**Authors:** Seri Jeong, Nuri Lee, Yeeun Park, Jaehong Kim, Kibum Jeon, Min-Jeong Park, Wonkeun Song

**Affiliations:** 1Department of Laboratory Medicine, Kangnam Sacred Heart Hospital, Hallym University College of Medicine, 1, Singil-ro, Yeongdeungpo-gu, Seoul 07441, Korea; hehebox73@hallym.or.kr (S.J.); nurilee822@hallym.or.kr (N.L.); yegrace91@hallym.or.kr (Y.P.); kim15198@hallym.or.kr (J.K.); mjpark@hallym.or.kr (M.-J.P.); 2Department of Laboratory Medicine, Hangang Sacred Heart Hospital, Hallym University College of Medicine, 12, Beodeunaru-ro, 7-gil, Yeongdeungpo-gu, Seoul 07247, Korea; pourmythe45@hallym.or.kr

**Keywords:** COVID-19, coinfection, mortality, prevalence, resistance, SARS-CoV-2

## Abstract

Coinfection rates with other pathogens in coronavirus disease 2019 (COVID-19) varied during the pandemic. We assessed the latest prevalence of coinfection with viruses, bacteria, and fungi in COVID-19 patients for more than one year and its impact on mortality. A total of 436 samples were collected between August 2020 and October 2021. Multiplex real-time PCR, culture, and antimicrobial susceptibility testing were performed to detect pathogens. The coinfection rate of respiratory viruses in COVID-19 patients was 1.4%. Meanwhile, the rates of bacteria and fungi were 52.6% and 10.5% in hospitalized COVID-19 patients, respectively. Respiratory syncytial virus, rhinovirus, *Acinetobacter baumannii*, *Escherichia coli*, *Pseudomonas aeruginosa*, and *Candida albicans* were the most commonly detected pathogens. Ninety percent of isolated *A. baumannii* was non-susceptible to carbapenem. Based on a multivariate analysis, coinfection (odds ratio [OR] = 6.095), older age (OR = 1.089), and elevated lactate dehydrogenase (OR = 1.006) were risk factors for mortality as a critical outcome. In particular, coinfection with bacteria (OR = 11.250), resistant pathogens (OR = 11.667), and infection with multiple pathogens (OR = 10.667) were significantly related to death. Screening and monitoring of coinfection in COVID-19 patients, especially for hospitalized patients during the pandemic, are beneficial for better management and survival.

## 1. Introduction

The coronavirus disease 2019 (COVID-19) pandemic, caused by severe acute respiratory syndrome coronavirus 2 (SARS-CoV-2), continues to be a serious and critical threat to global public health. The World Health Organization (WHO) reported 267,184,623 confirmed cases of COVID-19 and 5,277,327 deaths up to 9 December 2021 worldwide [1]. In South Korea, COVID-19 was first identified on 20 January 2020 [2]. SARS-CoV-2 has spread through several outbreaks, resulting in 503,606 confirmed cases and 4130 deaths up to 9 December 2021, in South Korea [3].

A variety of clinical manifestations in patients with COVID-19, from asymptomatic to severe illness and death, have been reported [4]. The common symptoms of COVID-19, such as fever, cough, and dyspnea, are indistinguishable from those caused by other respiratory pathogens [5]. Coinfection rates with other pathogens vary according to the geographic region and sampling period [6]. In patients with severe COVID-19, 0% viral coinfections were reported in Italy and China between December 2019 and March 2020 [7,8]. However, 57% of cases of viral coinfection were observed among 307 Chinese patients with COVID-19 between January and February 2020 [9]. Regarding bacterial coinfection, only 1% of coinfections were observed in hospitalized patients in the United Kingdom (UK) [10], whereas 74% of infections with bacterial pathogens were reported in patients with COVID-19 hospitalized during the New York City pandemic surge [11]. The fungal coinfection rates also varied from 1% [12] to 27% [13].

The major pathogens were influenza virus A/B (Flu A/Flu B), respiratory syncytial virus (RSV), rhinovirus (HRV), non-SARS-CoV-2 coronavirus strains, and adenovirus for viral coinfection. *Staphylococcus aureus*, *Haemophilus influenzae*, *Acinetobacter* species, *Escherichia coli*, *Klebsiella pneumoniae*, *Streptococcus pneumoniae*, and *Pseudomonas* species were the frequently isolated bacteria. Regarding fungi, *Candida* species and *Aspergillus* have mostly been reported [14]. Despite the limited number of studies investigating coinfection, respiratory pathogens have been reported to be associated with clinical outcomes in patients with SARS-CoV-2 infection [14]. Therefore, a study reflecting the recent prevalence of coinfection in patients with COVID-19, including specific types of pathogens, is required to provide better management during the pandemic.

In this study, we assessed the latest prevalence of coinfection in patients with COVID-19 for >1 year. The detailed characteristics, including commonly detected viruses, bacteria showing resistance, fungal species, laboratory findings, monthly timing of isolation, and risk factors related to mortality as a critical clinical outcome, were investigated.

## 2. Materials and Methods

### 2.1. Patients and Sample Collection

All samples from patients who requested SARS-CoV-2 testing and were diagnosed with COVID-19 were consecutively collected. Samples satisfying the amount required for testing for coinfection were included. A total of 436 samples from 57 inpatients (from August 2020 to October 2021) and 379 outpatients (from January 2021 to October 2021) were included. Nasopharyngeal swabs in the universal transport medium were obtained and immediately subjected to qualitative real-time reverse transcription polymerase chain reaction (RT-PCR) for detection of SARS-CoV-2. Viral ribonucleic acid (RNA) was extracted from swab samples, and RT-PCR was conducted targeting the envelope protein (E) gene and RNA-dependent RNA polymerase (RdRP) gene using the MagNA Pure 96 system (Roche Diagnostics, Rotkreuz, Switzerland) and STANDARD M nCoV Real-Time Detection kit (SD BIOSENSOR Inc., Gyeonggi-do, South Korea), respectively. RT-PCR was performed on a CFX96 Dx real-time PCR cycler (Bio-Rad Laboratories Inc., Hercules, CA, USA) according to the manufacturer’s instructions. When the samples were determined to be positive for SARS-CoV-2, the extracted nucleic acids were stored at −70 °C for further experiments. The basic demographics, clinical presentations, laboratory findings related to infection, and outcome data (such as in-hospital mortality data) were obtained after reviewing medical records.

### 2.2. Detection of Respiratory Viruses

The first obtained sample from a patient was used for the detection of coinfection of the virus. Respiratory virus multiplex real-time RT-PCR was used to concurrently detect 16 types of respiratory viral pathogens. The Allplex Respiratory Panel (Seegene Inc., Seoul, South Korea) was composed of three panels. Panel 1 included influenza subtypes A and B (Flu A and Flu B) and RSV subtypes A and B. Panel 2 targeted adenovirus, enterovirus, parainfluenza viruses 1/2/3/4, and metapneumovirus. Panel 3 could detect HRV, human bocavirus (HBoV) 1/2/3/4, and other human coronaviruses (229E/NL63/OC43). The MagNA Pure 96 system (Roche Diagnostics, Rotkreuz, Switzerland) and CFX96 Dx real-time RT-PCR cycler (Bio-Rad Laboratories Inc.) for RT-PCR reactions were also used as recommended by the manufacturer. Interpretation for respiratory virus positivity was automatically determined using Seegene Viewer software (Seegene Inc.).

### 2.3. Detection of Bacteria and Fungi

Clinical samples taken from inpatients with suspected infections and surveillance samples for epidemiological and infection control purposes were included. The first isolate of a given species recovered during the admission period for each patient was included. Bacteria were identified using the Vitek 2 system (bioMérieux, Hazelwood, MO, USA), and matrix-assisted laser desorption ionization-time-of-flight mass spectrometry using the Vitek-MS instrument (bioMérieux) was used for identification, as previously described [15]. Clinical information, including data on specimen type, sampling date, and hospitalization in the intensive care unit (ICU), was collected from medical records.

In terms of antimicrobial susceptibility testing, minimal inhibitory concentrations (MICs) of the detected pathogens were determined using the Vitek 2 system (bioMérieux). The tested antimicrobial agents included beta-lactam antibiotics, carbapenems, and cefoxitin for carbapenem-resistant *A. baumannii* (CRAB), carbapenem-resistant *P. aeruginosa* (CRPA), methicillin-resistant *S. aureus* (MRSA), methicillin-resistant coagulase-negative *Staphylococcus*, and extended-spectrum beta-lactamase (ESBL)-producing *Enterobacteriaceae*. MIC breakpoints for determining susceptibility were applied according to the Clinical and Laboratory Standards Institute guidelines (M100S) [16] and the European Committee for Antimicrobial Susceptibility Testing [17].

### 2.4. Statistical Analysis

Statistical analyses were performed using Analyse-it Method Evaluation Edition software, version 2.26 (Analyse-it Software Ltd., Leeds, UK) and MedCalc software, version 19.8 (MedCalc Software Ltd., Ostend, Belgium). Pearson’s chi-square tests for categorical variables were used for comparison. The Mann–Whitney *U* test was used for the assessment of continuous values. Non-normally distributed variables are presented as the median and interquartile range. To prevent confounding factors, univariate and multivariate binary logistic regression analyses were simultaneously used to examine the variables that correlated independently with the in-hospital mortality of patients with COVID-19. Coinfection was defined as a positive result for more than one virus, bacterium, or fungus in patients having positivity for SARS-CoV-2. In addition, the non-susceptibility rates of bacterial pathogens were calculated by dividing the number of isolates presenting non-susceptibility by the total number of isolates. All tests were two-sided, and differences with *p* values < 0.05 were considered statistically significant.

## 3. Results

### 3.1. Characteristics of Patients with COVID-19

The basic characteristics and laboratory findings of the 436 patients are summarized in Table 1. These patients with COVID-19 were classified into two subgroups according to their coinfection status. Infecting pathogens were detected in 8.7% (35/401) of all samples. The median age of patients with coinfection was significantly higher than that of patients with a single SARS-CoV-2 infection (*p* < 0.001). More than half of the patients with coinfection (65.7%) were >60 years old, whereas 40.9% patients with SARS-CoV-2 infection only were 40–59 years old. In terms of hospitalization status, 85.7% of patients with coinfection were inpatients. However, 93.3% of patients with a single SARS-CoV-2 infection were outpatients, showing a significant difference in distribution (*p* < 0.001). The median cycle threshold value of SARS-CoV-2 testing was approximately 20, and that of patients with coinfection was slightly elevated (*p* = 0.045 for the E gene and *p* = 0.046 for the RdRP gene). Among laboratory findings related to infection, the median procalcitonin level was significantly higher in patients with coinfection than in patients with a single SARS-CoV-2 infection (*p* < 0.001). In addition, C-reactive protein (*p* = 0.019) and lactate dehydrogenase (LD) (*p* = 0.024) levels were significantly elevated in the group with coinfection than in the group with a single SARS-CoV-2 infection. Relatively decreased lymphocytes were observed in patients with coinfection compared to those in patients with a single SARS-CoV-2 infection (*p* = 0.033).

### 3.2. Coinfection with Respiratory Viruses

The coinfection rate of respiratory viruses in patients with COVID-19 was 1.4% (6/436). The detected coinfecting viruses consisted of two respiratory syncytial viruses, two rhinoviruses, one bocavirus, and one influenza B virus. All patients presented with symptoms such as fever, dyspnea, and cough. Four patients had comorbidities, including hypertension. Among the six patients, two were hospitalized, and their chest X-ray images at admission revealed pneumonia with the presence of infiltrates. One of them developed dyspnea and had low saturation, which required mechanical ventilation and hemodialysis because of renal failure. After aggravation of acidosis, the patient died within 48 h. The monthly numbers of specific virus detections are shown in Figure 1. Further, 50% of viruses were detected between March and May. Regarding age groups, 50% of coinfecting viruses were identified in the 20–39 age group (Figure 2).

### 3.3. Nosocomial Infection with Bacteria

Among hospitalized patients with COVID-19, 52.6% (30/57) were infected with bacterial pathogens. More than half of the inpatients (57.0%) were transferred from other hospitals because of aggravation of dyspnea or pneumonia. The bacterial isolates from respiratory specimens, such as sputum, nasal swab, and tracheal aspiration, included 10 *A. baumannii*, 4 *P. aeruginosa*, 4 *S. aureus*, 3 *K. pneumonia*, and 1 *E. coli* strain. Among them, 9 of 10 *A. baumannii*, 3 of 4 *S. aureus*, 2 of 4 *P. aeruginosa*, and 1 of 3 *K. pneumoniae* strains were CRAB, MRSA, CRPA, and ESBL-producing *Enterobacteriaceae*, respectively. In terms of clinical outcome, 16 of 30 patients with COVID-19 required mechanical ventilation, and most of them died (93.8%). More than half of the patients (56.7%) were transferred from other hospitals because of the aggravation of dyspnea or pneumonia. The numbers of specific bacterial pathogens isolated monthly are shown in Figure 3. The detection rates in 2021 were higher than those in 2020. More than three bacteria causing nosocomial infection were isolated from January to April, July to August, and October 2021, coinciding with the pandemic periods in South Korea. Figure 4 shows the six bacterial pathogens detected in more than two isolates according to age group. The number of isolates detected in the >60 age group (70.3%) was higher than that in the 40–59 age group (29.7%).

### 3.4. Nosocomial Infection with Fungus

The fungal infection rate in hospitalized patients with COVID-19 was 10.5% (6/57). The infecting fungi consisted of four *C. albicans*, one *C. parapsilosis*, and one *C. tropicalis* strain. Among them, four *C. albicans* strains were isolated from sputum samples. Further, 67.0% of patients were transferred from other hospitals because of aggravation. All four patients with infection with CRAB or CRPA required mechanical ventilation and died. During spring and fall, the numbers of isolated fungi were evenly dispersed (Figure 5). The infecting fungi were mostly detected in the >60 age group (66.7%) (Figure 6).

### 3.5. Risk Factors for Mortality in Patients with COVID-19

The risk factors for mortality in patients with COVID-19 are shown in Table 2. According to logistic univariate analyses, older age (odds ratio [OR] = 1.060; *p* = 0.010), coinfection with more than one virus, bacteria, or fungi (OR = 13.818; *p* < 0.001), neutrophil count (OR = 1.077; *p* = 0.035), lymphocyte count (OR = 0.905; *p* = 0.042), and LD level (OR = 1.005; *p* = 0.022) were associated with mortality. In subgroup analyses of coinfection, bacterial infection (OR = 11.250; *p* < 0.001), infection with resistant bacteria (OR = 11.667; *p* = 0.009), and infections with more than two pathogens (OR = 10.667; *p* = 0.001) were significantly associated with mortality.

Multivariate analysis was performed with mortality as the binary dependent variable and age, coinfection, neutrophil count, lymphocyte count, and LD level as predictors. *p*-values < 0.05 were included in multivariate analysis. Subgroups of coinfection were not included because of multicollinearity with the coinfection variable itself. Independent associations between age (OR = 1.089; *p* = 0.015), coinfection (OR = 6.095; *p* = 0.033), and LD level (OR = 1.006; *p* = 0.041) and mortality were found.

## 4. Discussion

In this study, the latest prevalence of coinfection with a virus (1.4%), bacterium (52.6%) with 46.9% resistance, and fungus (10.5%) in patients with COVID-19 was investigated for >1 year of pandemic period between August 2020 and October 2021. The risk factors for mortality as a critical outcome in hospitalized patients were also analyzed, and significant associations between mortality and coinfection, age, and LD level were observed.

The isolated rates of viral coinfection in the present study (1.4%) were lower than those reported in previous meta-analyses (10% in a meta-analysis published in 2021 [14] and 3% in a meta-analysis published in 2020 [18]). The aggressive strategy of containment to prevent the spread of SARS-CoV-2 in South Korea during this study period could explain this low value. A previous study revealed the coinfection rates in 504 samples collected from March to July 2020 from four Korean hospitals [19]. The values ranged from 1.3% to 3.3%, which is similar to that reported in our study. Moreover, a study including 342 patients with COVID-19 demonstrated a viral coinfection of 7.9% in South Korea [20]. Locoregional situations such as an outbreak in Kyungpook province and sampling timing limited to February 2020 might have caused this difference. Previous studies in Korea reported only the prevalence of coinfection rates. However, the association between coinfection and clinical outcomes in hospitalized patients has not been presented. The increasing prevalence of infections caused by multidrug-resistant bacteria, as well as viruses and fungi, during the COVID-19 pandemic would become a threat to public health in South Korea. Simultaneous testing for coinfection with other pathogens is required considering the increase in coinfection during COVID-19 outbreaks. Respiratory symptoms in COVID-19 may contribute to increased infectivity through the dispersal of aerosols. Therefore, a study with a longer period covering all seasons is important to reflect the current status of coinfection. The low rate of viral coinfection seemed to be maintained in 2021, similar to that in 2020 under containment, with measures such as social distancing and active mask wearing. Concordant with the results of previous studies [14,18,21], RSV, HRV, and influenza were commonly detected in this study. Coinfection with RSV is associated with a longer length of stay [22]. HRV with a long period of viral shedding and environmental resistance [23] has been associated with higher morbidity rather than mortality, causing an economic burden [24]. HBoV was also observed in a previous study using the same RT-PCR kit in South Korea [20]. HBoV, an enveloped virus, was detected consistently during the COVID-19 pandemic according to the Korea Influenza and Respiratory Viruses Surveillance System [23]. The higher detection of viral pathogens in the 20–39 age group suggests that young patients with COVID-19 are more susceptible to respiratory viral coinfection, consistent with the findings of a previous study [20].

The nosocomial infection rate caused by bacteria in patients with COVID-19 was 6.9%, and that in critically ill patients was higher (8.1%) according to a meta-analysis [25]. Our data showed a nosocomial infection rate by bacteria of 52.6%, which was higher than the rates reported in other studies. Most of our patients were hospitalized in the ICU (98.2%) and transferred from other hospitals with severe symptoms and longer hospitalization days. Furthermore, a relatively small number of patients were included because culture for infection with bacterial pathogens was conducted only for inpatients. These factors could contribute to the high nosocomial infection rates observed in the present study. A previous study that included hospitalized patients with COVID-19 during the New York City pandemic surge revealed a bacterial infection of 74% within 30 days of admission [11]. Studies performed in France (47.5%) [26] and Spain (38.6%) [27], including patients admitted to the ICU, also showed a high prevalence of bacterial infection. Concordant with our study, study populations such as critically ill patients with pneumonia and immunocompromised state due to COVID-19 treatment could be the cause of this prevalence. The most commonly isolated bacteria were *A. baumannii*, *E. coli*, *P. aeruginosa*, *S. aureus*, and *K. pneumoniae*. These species have been frequently reported as pathogens of infection in previous studies [14,18,28]. *A. baumannii* was mainly isolated from respiratory samples 48 h after admission. This species is the predominant pathogen in patients with ventilator-associated pneumonia [29]. *P. aeruginosa*, *S. aureus*, and *K. pneumoniae* also cause respiratory tract infections [28]. Regarding antimicrobial resistance, the rates varied from 33.3% for ESBL-producing *K. pneumoniae* to 90.0% for CRAB according to the infecting species. CRAB, which showed the highest resistance rate in our study, is an opportunistic pathogen associated with nosocomial infections [30]. It affects critically ill patients requiring mechanical ventilation in the ICU and is the leading cause of mortality due to infection [31]. During the COVID-19 pandemic, an increased risk of carbapenem-resistant infections has been reported [32]. The increasing trends could be attributed to excessive use of antimicrobial agents and vulnerability of patients with COVID-19 due to SARS-CoV-2-associated immune dysfunction and prolonged hospitalization [33]. These data emphasize adherence to infection control practices, such as hand hygiene and appropriate use of antibiotics to prevent transmission [34]. In terms of multiple bacterial infections during the COVID-19 pandemic, aerosol generation due to symptoms of COVID-19 and the immunocompromised state of severe patients with COVID-19 with multiple comorbidities receiving multiple treatments, including steroids, might enhance serious infections [6,28]. Regarding age groups, bacterial infection frequently occurs in elderly populations. The median age of the study population for bacterial infections in patients with COVID-19 hospitalized during the New York City pandemic surge was 62 years. Furthermore, the median age of the study population in several studies [25,35] for bacterial infection in patients hospitalized in the ICU was >60 years, concordant with our study results, revealing higher bacterial pathogens in the elderly population.

Fungal infection cases increased during the COVID 19 pandemic [36,37]. The prevalence of fungal infection was 10.5%, based on our data, including patients admitted to the ICU requiring ventilation. The most commonly detected isolate was *C. albicans* from respiratory specimens, consistent with the findings of other studies [13,37]. Impaired immune responses due to infection with SARS-CoV-2 and the use of immunosuppressants, including steroids and tocilizumab, for the treatment of COVID-19 could be the cause of increased fungal infection [36,37]. Early detection of fungal infection and antifungal therapy is associated with better survival [13]. Therefore, simultaneous testing for fungal infection and antifungal prophylaxis is required for critically ill patients with COVID-19.

Coinfection with other pathogens, aging, and elevated LD levels are risk factors for mortality in hospitalized patients with COVID-19. In particular, nosocomial infection with bacteria has been reported to be lethal in patients admitted to the ICU [38,39]. In the present study, we demonstrated the independent association of coinfection through multivariate analysis, compensating for confounding factors. Coinfection in patients with COVID-19 is an important factor because of its modifiability via interventions for infection control. Concurrent screening for other pathogens with appropriate therapy as well as maintaining distancing, enhanced monitoring of hand hygiene and antibiotic prescription, and use of personal protective equipment improve clinical outcomes. Older age has been an inevitable risk factor in other studies [40,41], which is consistent with our study’s findings. Severe medical comorbidities and decreased immune responses may affect outcomes. A previous study showed that an elevated serum LD level was a prognostic indicator in patients with COVID-19 [42]. LD abnormality is a consequence of hypoxia and tissue injuries caused by inflammation. However, the non-specificity of LD makes us focus on coinfection as a modifiable factor with a high odds ratio (6.1) for mortality.

Our study has the following limitations. First, our study included patients visiting a single university hospital with a relatively small study population. Further studies including a large number of patients from multi-centers are necessary to verify the actual prevalence of coinfection and confirm the risk factors. Second, viruses were detected using commercially available multiplex RT-PCR assays. Other viruses that were not targeted by this assay could not be detected, leading to a lower coinfection rate. Third, bacterial and fungal identification were performed only in inpatients with COVID-19 because tests for these pathogens were not broadly requested for outpatients. Additional assessment of bacterial and fungal infections in these patients is required to determine the precise prevalence of nosocomial infection and its impact on clinical outcomes.

## 5. Conclusions

In the present study, we assessed the recent prevalence of coinfection with viruses, bacteria, and fungi in patients with COVID-19 over a period of >1 year to reflect seasonal variations in pathogens. The rate of viral coinfection was maintained at a low value owing to the aggressive containment strategy to prevent the spread of COVID-19. Moreover, the rates of nosocomial infection by bacteria and viral coinfection in critically ill patients, mostly in the ICU, were high. The risk factors for mortality in patients with COVID-19 as a critical outcome were analyzed, and infection, especially with bacteria, was significantly related to mortality. In addition, infection with resistant bacteria and multiple pathogens was also associated with mortality in patients with COVID-19. This study included microbial pathogens, including viruses, bacteria with multidrug resistance, and fungi concurrently, which has been limitedly reported in patients with COVID-19 during th long pandemic. To the best of our knowledge, this is the first study on an association between coinfection and clinical outcomes in hospitalized South Korean patients. Based on our data, screening and monitoring of coinfection in patients with COVID-19, particularly for hospitalized patients during the pandemic, are recommended for better management and outcomes.

## Figures and Tables

**Figure 1 viruses-14-00446-f001:**
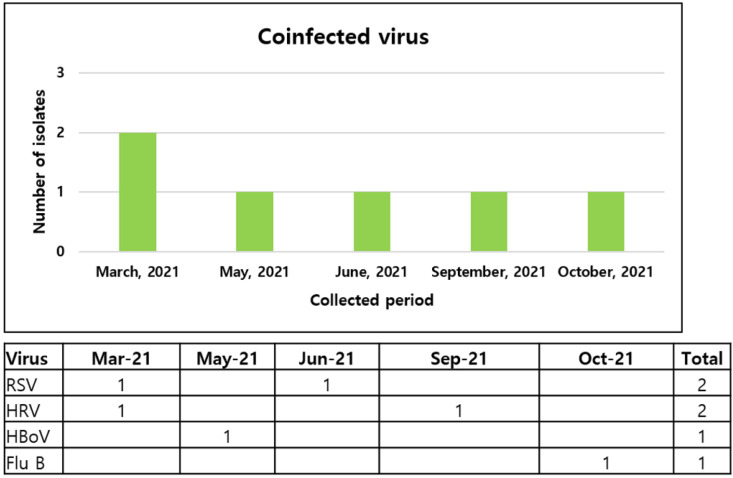
Monthly detection of coinfecting viruses in patients with COVID-19. Flu B, influenza B; HBoV, bocavirus; HRV, rhinovirus; RSV, respiratory syncytial virus.

**Figure 2 viruses-14-00446-f002:**
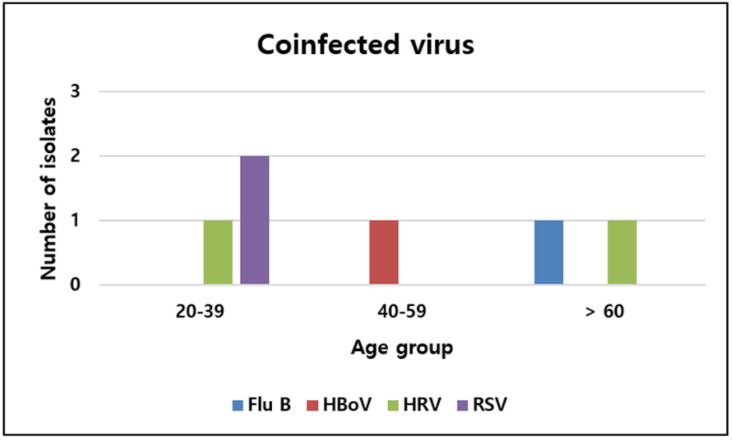
Distribution of coinfecting viruses by age group in patients with COVID-19. Flu B, influenza B; HBoV, bocavirus; HRV, rhinovirus; RSV, respiratory syncytial virus.

**Figure 3 viruses-14-00446-f003:**
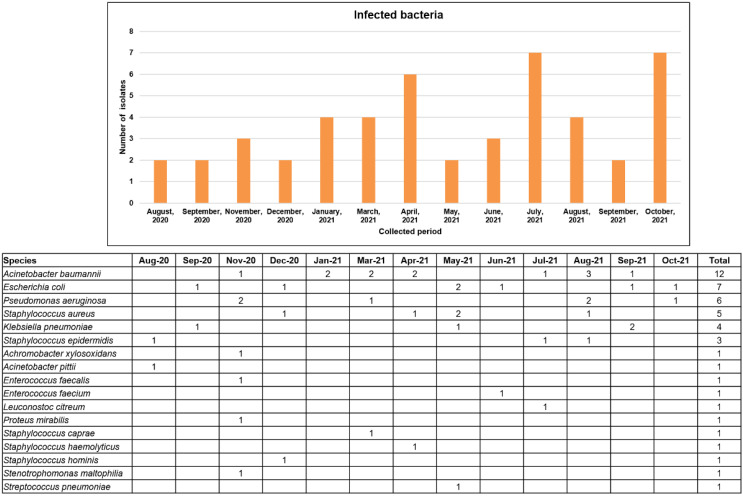
Monthly detected bacteria causing nosocomial infection in patients with COVID-19.

**Figure 4 viruses-14-00446-f004:**
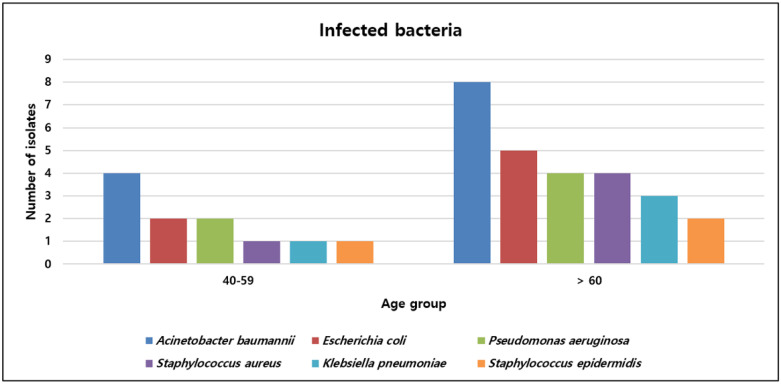
Distribution of bacteria causing nosocomial infection by age group in patients with COVID-19.

**Figure 5 viruses-14-00446-f005:**
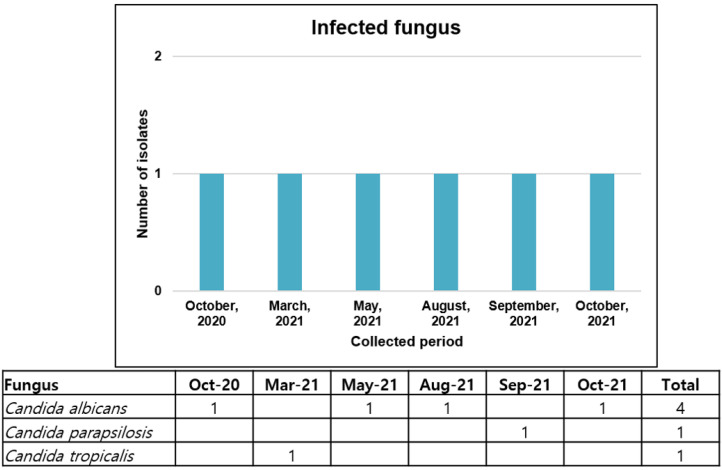
Monthly detected fungi causing nosocomial infection in patients with COVID-19.

**Figure 6 viruses-14-00446-f006:**
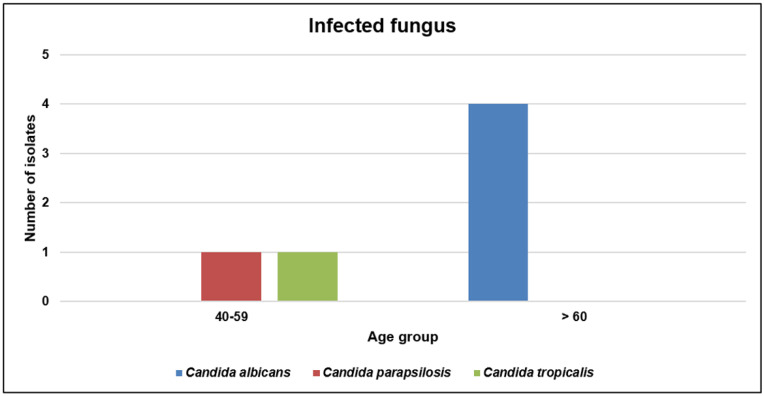
Distribution of infecting fungi by age group in patients with COVID-19.

**Table 1 viruses-14-00446-t001:** Characteristics of patients with COVID-19 *.

Characteristics	Total (*n* = 436)	Coinfection (*n* = 35)	SARS-CoV-2 Only (*n* = 401)	*p* ^†^
Median age, years	48.0 (31.0–59.0)	70.0 (56.2–77.0)	46.0 (30.0–58.0)	<0.001
Age group				
<20 years	37 (8.5)	0 (0.0)	37 (9.2)	<0.001
20–39 years	120 (27.5)	3 (8.6)	117 (29.2)	
40–59 years	173 (39.7)	9 (25.7)	164 (40.9)	
>60 years	106 (24.3)	23 (65.7)	83 (20.7)	
Sex				
Male	220 (50.5)	18 (51.4)	202 (50.4)	0.905
Female	216 (49.5)	17 (48.6)	199 (49.6)	
Hospitalization				
Inpatient	57 (13.1)	30 (85.7)	27 (6.7)	<0.001
Outpatient	379 (86.9)	5 (14.3)	374 (93.3)	
Ct value				
E gene	20.6 (15.3–29.0)	24.3 (20.1–28.0)	19.9 (15.1–29.1)	0.045
RdRP gene	20.5 (15.0–29.0)	24.1 (19.8–27.6)	20.1 (14.6–29.1)	0.046
Laboratory finding				
Procalcitonin (ng/mL)	0.2 (0.1–0.9)	0.3 (0.1–3.9)	0.1 (0.0–0.1)	<0.001
CRP (mg/L)	64.3 (20.9–139.7)	117.7 (47.4–151.0)	35.3 (15.0–100.7)	0.019
WBC (10^9^/L)	8.5 (5.6–11.3)	9.0 (5.5–14.8)	7.8 (5.8–8.9)	0.051
Neutrophil (%)	86.6 (78.5–91.7)	88.7 (81.5–93.0)	83.5 (76.2–89.2)	0.055
Lymphocyte (%)	7.0 (4.5–14.1)	6.0 (3.0–12.0)	10.2 (5.3–16.8)	0.033
Glucose (mg/dL)	157.5 (126.3–200.4)	147.0 (117.0–204.2)	158.0 (128.5–195.3)	0.602
LD (IU/L)	444.5 (320.8–527.0)	493.0 (366.7–547.3)	358.0 (310.0–495.2)	0.024

* Data are expressed as the median (1st to 3rd quartile) for continuous variables and the number (percentage) for categorical variables. ^†^
*p* values were calculated for the differences between Coinfection and SARS-CoV-2 only. Ct, cycle threshold; CRP, C-reactive protein; WBC, white blood cell; LD, lactate dehydrogenase.

**Table 2 viruses-14-00446-t002:** Univariate and multivariate analyses of inpatients with COVID-19 for mortality.

Variable	Univariate		Multivariate *	
	OR (95% CI)	*p*	OR (95% CI)	*p*
Age	1.060 (1.014–1.107)	0.010	1.089 (1.017–1.166)	0.015
Sex ^‡^				
Male	Reference			
Female	0.500 (0.158–1.583)	0.239		
Comorbidity ^‡^	0.661 (0.270–2.293)	0.787		
HTN	0.818 (0.252–2.660)	0.739		
DM	0.978 (0.889–1.076)	0.646		
Infection with other pathogens ^‡^	13.818 (3.369–56.678)	<0.001	6.095 (1.160–32.028)	0.033
Coinfection, virus	NA ^†^	0.998		
Nosocomial infection, bacteria	11.250 (3.041–41.624)	<0.001		
Nosocomial infection, resistant bacteria	11.667 (1.863–73.068)	0.009		
Nosocomial infection, fungus	2.605 (0.399–17.008)	0.317		
Multiple coinfection	10.667 (2.505–45.419)	0.001		
Laboratory finding				
Procalcitonin (ng/mL)	0.998 (0.983–1.013)	0.797		
CRP (mg/L)	1.005 (0.998–1.012)	0.135		
WBC (10^9^/L)	1.097 (0.979–1.228)	0.111		
Neutrophil (%)	1.077 (1.005–1.154)	0.035	1.115 (0.970–1.282)	0.125
Lymphocyte (%)	0.905 (0.821–0.996)	0.042	1.072 (0.883–1.302)	0.480
Glucose (mg/dL)	1.002 (0.996–1.007)	0.544		
LD (IU/L)	1.005 (1.001–1.010)	0.022	1.006 (1.000–1.012)	0.041

* Variables less than 0.05 of the *p* value in univariate analysis were included in the multivariate analysis. ^†^ Odds ratio for virus coinfection was not applicable because only one hospitalized patient was coinfected with the virus. ^‡^ The numbers of patients for categorical variables were as follows: male (*n* = 36), female (*n* = 21), HTN (*n* = 17), DM (*n* = 11), nosocomial infection, bacteria (*n* = 28), nosocomial infection, resistant bacteria (*n* = 18), nosocomial infection, fungus (*n* = 5), and multiple coinfection (*n* = 14). OR, odds ratio; CI, confidence interval; HTN, hypertension; DM, diabetes mellitus; NA, not applicable; CRP, C-reactive protein; WBC, white blood cell; LD, lactate dehydrogenase.

## Data Availability

All data used and presented in this study was deposited in https://dataverse.harvard.edu/ (https://doi.org/10.7910/DVN/7OWB4W), accessed on 20 December 2021.

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
