# Peer review of "Prevalence and Clinical Impact of Coinfection in Patients with Coronavirus Disease 2019 in Korea"

_viruses, 2022, doi:10.3390/v14020446_

Round 1
Reviewer 1 Report
In this study, the authors examined the severity of co-infections with SARS-CoV-2 and respiratory pathogens in COVID-19 patients for >1 year. But readers feel frustrated when this work was conducted in only a very small number of patients.
This study does not provide any new information, because many studies in this topic have been published and performed on a very large sample size.
The relationship between the number of patients and the number of respiratory samples examined is not reported.
What are the selection criteria for samples? The authors mentioned that "All samples from patients who requested SARS-CoV-2 testing and were diagnosed 66 with COVID-19 were consecutively collected". But only 436 samples from 57 inpatients and 379 outpatients were collected during one year of study.
The definition of co-infection is not clear in my opinion: Coinfection was defined as a positive result for more than one virus, bacteria, or fungus. The authors included also the co-infected patients with multiple respiratory pathogens but negative for SARS-CoV-2.
Were multiple samples tested for each subject? This aspect can help discriminate “co-infections” from “super-infections”.
In my opinion, the bacterial co-infection samples are due to superinfection, or nosocomial infection, not true "co-infection". Because there are in hospitalized patients.
The analyzes of hospitalized patients with COVID-19 for mortality doesn't really make sense, because there are only 57 patients in total with 30 co-infections
Reviewer 2 Report
Interesting study especially in the era of SARS-CoV-2. Studies are starting to indicate the impact of co-infections between SARS-CoV-2 and other respiratory pathogens, and their association with severe disease and poor outcomes.
Reviewer 3 Report
The issue of coinfection in patients with COVID-19 infection is of interest for clinical practice. However, it is clear that there are local factors that affect the germs that are detected.
For this reason, the article, which is of interest, should reflect its local character.
First of all, the title should state that it is an article referring to Korea. It would also be interesting to emphasize this local factor in the text.
The tables and figures should be improved, as they should have a homogeneous style. In this sense, I would eliminate tables 2 and 3 which, although they offer detailed information on the patients, seem to me not to be of interest to the reader.
Once these aspects have been modified, the text could be re-evaluated for publication.
Round 2
Reviewer 1 Report
I suggest adding the number of patients in Table 2 and changing the term "Coinfection" (with OR = 13.818 (3.369-56.678)) as appropriate
Reviewer 3 Report
The new version is really improved so I will accept it to be published in the magazine in the present form.
Author Response
- We were very appreciated with your constructive comments for our manuscript. We made utmost efforts to address the comments.